# Parallel fast and slow recurrent cortical processing mediates target and distractor selection in visual search

Sarah E. Donohue [1,2,3], Mircea A. Schoenfeld[1,2,4] & Jens-Max Hopf [1,2 ✉]

Visual search has been commonly used to study the neural correlates of attentional allocation in space. Recent electrophysiological research has disentangled distractor processing from target processing, showing that these mechanisms appear to operate in parallel and show electric fields of opposite polarity. Nevertheless, the localization and exact nature of this activity is unknown. Here, using MEG in humans, we provide a spatiotemporal characterization of target and distractor processing in visual cortex. We demonstrate that source activity underlying target- and distractor-processing propagates in parallel as fast and slow sweep from higher to lower hierarchical levels in visual cortex. Importantly, the fast propagating target-related source activity bypasses intermediate levels to go directly to V1, and this V1 activity correlates with behavioral performance. These findings suggest that reentrant processing is important for both selection and attenuation of stimuli, and such processing operates in parallel feedback loops.

[1] Otto-von-Guericke University Magdeburg, 39120 Magdeburg, Germany. [2] Leibniz Institute for Neurobiology, 39118 Magdeburg, Germany. [3] University of Illinois College of Medicine Peoria, 61605 Peoria, IL, USA. [4] Kliniken Schmieder Heidelberg, 69117 Heidelberg, Germany. ✉email: jens-max.hopf@med.ovgu.de

Event-related potential (ERP) and event-related magnetic field (ERMF) recordings enable the assessment of dynamic brain activity underlying attentional selection with high temporal resolution. Over the last 25 years, one component of the ERP has featured prominently as a tool to investigate cortical mechanisms of visual attention: the N2pc[1,2]. The N2pc (**N2 p**osterior **c**ontralateral) is a negative voltage deflection of the ERP elicited over the occipital scalp contralateral to the visual field where a to-be-attended item is presented (cf. Fig. 1a). It is typically analyzed in visual search tasks where it appears roughly 200–300 ms after search frame onset. Numerous ERP studies in humans have shown that the N2pc reliably indexes the focusing of attention onto the target item[1–6], with the resolution of ambiguities of visual coding being a core mechanism[4,7,8]. Moreover, the N2pc is a universal index of attentional selection across species, as an N2pc analog (mN2pc) has been documented in the monkey[9–12].

Using current source localization analyses based on parallel ERP and ERMF recordings, the N2pc was found to be generated to a great extent in ventral extrastriate cortex areas, with a small early contribution from the parietal cortex[13]. Notably, the ventral extrastriate sources had no fixed cortical origin. They appeared at variable hierarchical levels depending on the spatial resolution required to discriminate the target among competing distractors[7]. Increasing the resolution of discrimination caused source activity to appear at progressively lower hierarchical levels of cortical representation, suggesting that activity underlying the N2pc propagates in the reverse hierarchical direction in the ventral extrastriate cortex to levels where receptive field size is at the spatial scale of stimulus separation, thereby allowing the resolution of the spatial competition among items[4,7,8,14].

More recent research revealed that the human N2pc consists of subcomponents (cf. Fig. 1b) with opposite-field polarity, which were called target negativity (Nt) and distractor positivity (Pd)[15]. There is now substantial experimental evidence that the Nt reflects target selection and the Pd reflects the attenuation of salient distractors[15–22]. Given this separation into subcomponents, it is currently unclear which component of the N2pc undergoes the backpropagation of activity in the visual cortex. If activity modulations at progressively lower levels index the attenuation of irrelevant input, the Pd should account for the backpropagation of activity. Alternatively, if it reflects a bias to enhance the target representation at progressively lower levels, the Nt should show backpropagation. Finally, it is possible that both subcomponents operate in a combined push-pull-like manner across the visual cortex hierarchy, with activity underlying both components showing backpropagation. At present, the cortical sources of the Nt and Pd are not characterized. A topographical analysis of the Nt and Pd field distribution revealed that the Nt appeared over more ventrolateral and the Pd over more dorsomedial cortex[15], leading to the hypothesis that the Pd reflects the parietal contribution to the N2pc while the Nt arises from the ventral extrastriate cortex. Other studies, in contrast, found the topographical field distributions of the Nt and Pd to be very similar, suggesting that both index a modulation process in the same cortical substrate with opposite polarity[18]. The major goal here is therefore twofold. We will first provide a full spatio-temporal characterization of current source activity generating the Nt and Pd and then aim at clarifying the dynamics of recurrent processing underlying these subcomponents in the visual cortex.

## Results

### Derivation of the target- and distractor-related response. 
The N2pc is typically derived as illustrated in Fig. 1a. While fixating

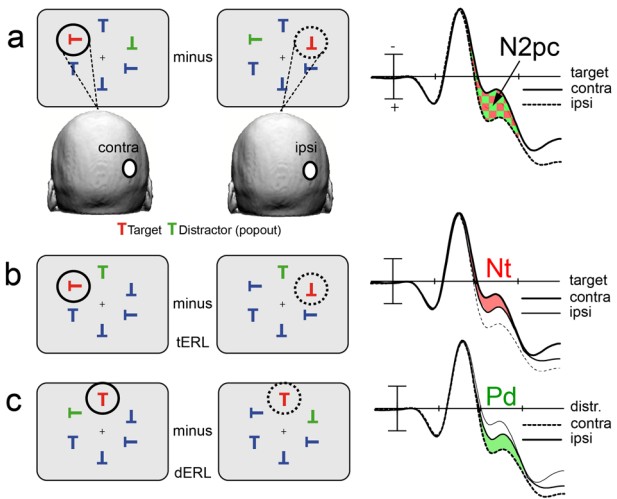

**Fig. 1 Design of search arrays and derivation of the N2pc, Nt, and Pd.**
**a** Balanced search arrays with the target (red T) and the distractor (green T) being presented at equivalent positions in opposite visual fields (VFs). The N2pc is derived at a given recording site (white circle over the right occipital cortex) by comparing the response elicited by a target (red T) in the contralateral (contra, solid black trace) versus the ipsilateral VF (ipsi, dashed black trace). The N2pc represents the contra-minus-ipsi difference shown as red/green area between curves. As the target is always paired with a distractor in the opposite VF, the contra-minus-ipsi difference will reflect the response to both the target (red) and the distractor (green). **b** The Nt (red area between traces) is isolated in the event-related lateralization (ERL) by presenting the target in the left or right VF while the distractor appears at the same vertical meridian position (target-related ERL) so that the response from the distractor is canceled in the contra-minus-ipsi difference. **c** Derivation of the Pd (green area between traces) in the distractor-related ERL (dERL) by presenting the distractor in the left versus right VF and the target at corresponding vertical meridian positions.

the center of a computer screen, subjects covertly attend an array of items, where a target item in one visual hemifield (the red T in the left VF) is presented together with a distractor in the opposite hemifield field (the green T in the right VF) among neutral distractors (blue Ts) in both hemifields. For a given electrode site (white circle), The N2pc is derived by subtracting the waveform elicited by a target presented ipsilateral to this electrode (dashed black trace) from the waveform elicited by a target contralateral (solid black trace). This N2pc difference (red/green area between waveforms) reflects hemisphere lateralized activity elicited by the target and the distractor singleton. The response from the neutral blue distractors is eliminated under the subtraction because those items elicit similar activity in both visual fields. To disentangle the response to the target singleton from that of the distractor singleton (henceforth referred to as target and distractor, respectively), asymmetric search arrays can be used[5] as illustrated in Fig. 1b and c. That is, to isolate the lateralized response to the target (target-elicited event-related lateralization, tERL, Fig. 1b), the response to the distractor can be nulled in the N2pc difference by placing it at the same vertical meridian position for left and right visual field targets. Analogously, to isolate the lateralized response to the distractor (distractor-elicited event-related lateralization, dERL, Fig. 1c), the response to the target can be nulled by placing it on a vertical meridian position. It was observed[15] that the lateralized target-elicited contralateral negativity, referred to as target negativity (Nt, red area between traces in Fig. 1b), which turned out to be consistent with the relative polarity of the N2pc. The lateralized distractor, in contrast, elicited a contralateral positivity, dubbed distractor positivity (Pd, green area

between traces in Fig. 1c). In symmetrical search arrays, as shown in Fig. 1a, the derivation of the N2pc (contra-minus-ipsi difference) obscures these subcomponents because the target and the distractor singleton appear in opposite visual fields so that the Pd adds as relative negativity to the Nt (red plus green area between traces)[20].

**The overall time course of the response elicited by a lateralized target and distractor**. Figure 2 shows the magnetic field response (solid lines) elicited by a lateralized singleton target (tERL, upper) and a lateralized singleton distractor (dERL, lower) at selected sensor sites over the left and right posterior-lateral cortex (see inset). The waveforms are collapsed over hemispheres and sensors measuring the maximum efflux (purple) and influx (blue) component of the field distribution. As will become clear below, the field distributions underlying the tERL and the dERL change topography over time. The shown waveforms at the selected sensors can therefore give only a rough impression of the time course of the magnetic field response.

Both the tERL and dERL display very similar initial modulation peaks (dark gray) reflecting the N1pc. We will not consider the N1pc further in this paper. An analysis of current sources underlying the N1pc, however, is reported in the Supplementary Materials (Supplementary Fig. 1 and Supplementary Movie 3). A detailed functional characterization of the N1pc is provided in ref. [23]. The N1pc is followed by a negative polarity modulation between 190 and 350 ms (light red), which represents the Nt component in the tERL. The Nt, in turn, is followed by a second negative modulation between 350 and 450 ms, which we refer to as the post-Nt (Fig. 2 dark gray). We will not focus on the time range of the post-Nt; however, a short discussion of the post-Nt is provided in the Supplementary Materials. In the dERL below, the N1pc is followed by a positive polarity modulation—the distractor positivity (Pd, light green). In contrast to the tERL, the dERL does not show modulation in the time range of the post-Nt. To verify time ranges of significant ERMF modulation underlying the tERL and dERL, one-way topographic analyses of variance (TANOVA)[24,25] with the factor-levels target or distractor in left versus right VF were computed for each time sample between 0 and 500 ms after stimulus onset (see "Methods" for details). The result is shown by the dashed traces below the magnetic waveforms, where significant time ranges are shown as grayed areas. The TANOVAs clearly confirm the presence of three significant time ranges corresponding with the N1pc, the Nt and the post-Nt for the tERL, and two significant time ranges corresponding with the N1pc and the Pd in the dERL. Note, the TANOVAs were computed based on all recorded sensors, the ranges of significant topographical variation do not perfectly match the component ranges highlighted under the waveforms, as they reflect the response at the selected sensors shown in the inset. The Nt and Pd represent opposite polarity modulations in both the electric and the magnetic field response. To keep terminology simple and consistent with the ERP components, we collapsed the magnetic response by subtracting the efflux from the influx field component, such that the magnetic Nt (N2pc) appears as negative polarity modulation.

In the sections below, we first provide a detailed description of the sequence of activity propagation in the parietal/visual cortices for the Nt and Pd. We then highlight the modulations that are of high interest, and we briefly discuss the potential functional role of such a sequence of activity. Finally, we compare the activity patterns of these two components.

**ERMF topography and source localization of the Nt**. Figure 3 shows field topographies and corresponding current density

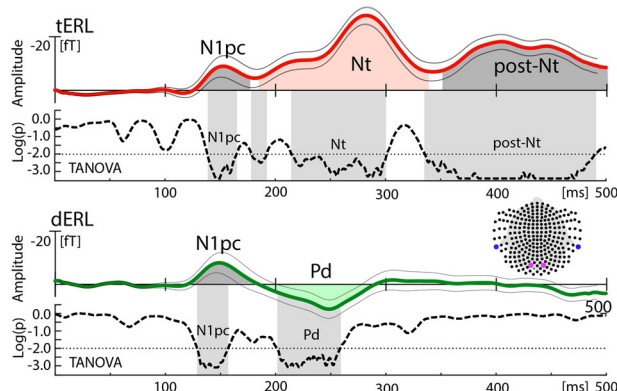

**Fig. 2 Overall time course and statistical validation of the event-related lateralization (ERL) elicited by the target (tERL) and the distractor (dERL).** The event-related magnetic field (ERMF) waveforms (colored solid traces) show the hemisphere-collapsed tERL (red) and dERL (green) responses recorded at the blue and purple sensor sites marked in the inset. The Nt and Pd component are highlighted by the red and green area under the curve, respectively. The dashed lines below the waveforms show the time course of statistical differences (log($p$)-values) determined with a time sample-by-sample topographical ANOVA (TANOVA, see "Methods"). Significant time ranges corrected for multiple comparisons (threshold shown by the dotted line) are highlighted by light-gray areas. The thin gray traces mark the standard error of the tERL and dERL.

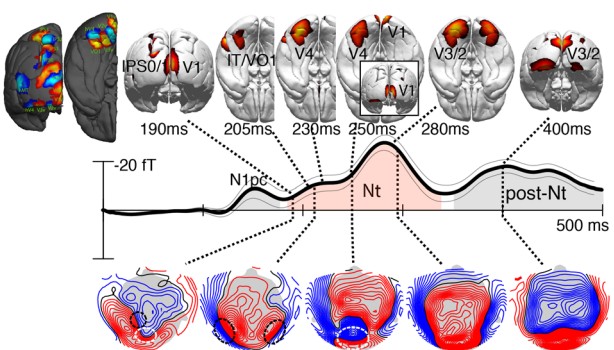

**Fig. 3 Event-related magnetic field (ERMF) distributions and current density reconstructions (CDR) of the Nt component.** The 3D-CDR maps show activity above an arbitrary threshold to illustrate the source maxima at representative time points during successive stages of the backpropagation of source activity in the visual cortex. The waveform replots the hemisphere-collapsed tERL response shown in Fig. 2 to illustrate the approximate latency of the shown CDR and ERMF distributions. The ERMF maps show the field distribution corresponding with the CDR maps. The field lines (blue/red) separated by increments of 2fT. The hot-scale and blue-scale areas rendered onto the dark gray hemispheres display probabilistic maps of retinotopic areas (see "Methods").

reconstruction (CDR) maps at subsequent time points in the Nt time range (red area under the curve). The initial field distribution of the Nt at ~190 ms shows an efflux–influx configuration (blue-red field lines) over the left posterior parietal cortex (black dashed ellipse), as well as over more posterior central areas (white ellipse). The corresponding CDR map shows that the underlying source maxima arises in left inferior parietal cortex areas IPS0/1/2 and in the early visual cortex. The left hemisphere parietal source contributing to the Nt in this early time range dovetails with a previous observation of an initial parietal modulation underlying the magnetic N2pc[13]. Also in line with previous analyses of the

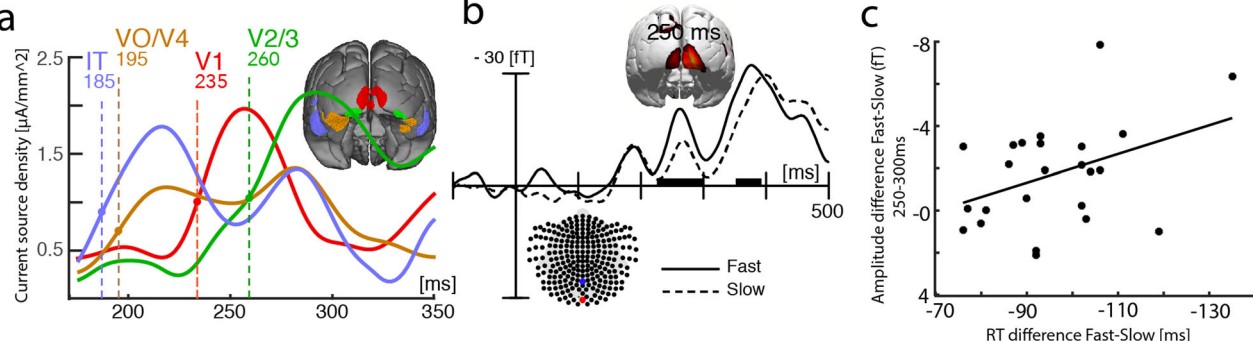

**Fig. 4 Source activity underlying the Nt at different hierarchical levels in the visual cortex. a** The source waves represent the current density reconstructions (CDR) in the shown regions of interest (ROIs, collapsed over regions in the left and right hemisphere) with the corresponding coloration. The response maximum in V1 (red) appears with a delay relative to the higher-level (blue, IT) and mid-level ROIs (brown, VO/V4), but notably prior to the maximum in lower-level areas (green, V2/3). The colored numbers indicate the time point (ms) at which the source strength in the corresponding ROIs reached 50% of the maximum CDR in the analyzed time range. **b** Event-related magnetic field (ERMF) response elicited by the target at occipital-central sensors (inset) reflecting activity in V1 for fast (solid) and slow (dashed) correct responses (median split). **c** Correlation (across $n = 24$ subjects) of the response time difference between fast and slow responses and the corresponding amplitude difference at the occipital sensors shown in (**b**).

magnetic analog of the N2pc[7,8,13,26], the parietal field pattern is followed by efflux–influx components over posterior-lateral regions in both hemispheres around 200 ms (black dashed ellipses). Corresponding source maxima appear in the higher-tier ventral extrastriate cortex (VO1 and higher, IT). Within ~45 ms, this anterior–lateral–ventral maximum propagates toward more posterior lower-tier areas passing through V4/V3 to reach V2. A movie illustrating the backward propagation of Nt source activity as well as the corresponding change of the ERMF distribution over time is provided in Supplementary Movie 1. Source activity in higher and mid-level ventral extrastriate areas is overall stronger in the left than the right hemisphere.

Importantly, around 230 ms, a strong occipital–polar field effect (dashed white ellipse) emerges transiently and disappears before the ventral back-propagating source activity reaches early visual areas (V3, V2) around 265 ms (max. at ~285 ms). The CDR map at 250 ms (inset) indicates that the underlying current source maximum locates to the primary visual cortex (V1). Figure 4a shows the time course (between 175 and 350 ms) of source activity (source waves) in cortical regions of interest (ROIs, collapsed over hemispheres) corresponding with high-level (blue, IT), high-to-mid level (brown, VO1, V4), low-level extrastriate cortex (green, V3/V2), as well as the primary visual cortex (red, V1). The temporal onset of source activity is indicated by the colored dashed lines, which mark the latency at which source activity in a given ROI approached 50% of its maximum in the analyzed time range. The initial source activity appears in anterior–lateral extrastriate areas around 185 ms (blue trace, max. ~215 ms), followed in high-mid-level areas around 195 ms (max. ~220 ms). This is, in turn, followed by a source maximum in the primary visual cortex arising around 235 ms (red trace, max. ~255 ms) ~30 ms prior to the onset of source in lower-level areas V3/2 (green) at 260 ms (max. ~290 ms). The functional implications of this pattern of activity are discussed below.

The observation that recurrent activity in V1 appears earlier than recurrent activity in V2/V3 is very notable insofar as it implies that the backpropagation of source activity is not a single process that steps down level by level within the cortical hierarchy. Instead, it suggests that the backpropagation involves parallel sweeps of fast and slow of recurrent processes, with the fast sweep directly reaching back to V1 prior to a slower level-by-level back-propagating modulation that reaches early visual cortex areas (V3, V2) with a delay. The observation raises the question about the functional role of the fast recurrent sweep to

V1. Recurrent activity in V1 has been suggested to enhance the representation of information at the input stage of processing to aid attention and awareness[27–30], where the spatial resolution is high to facilitate discrimination. If this would be the case, one would predict a stronger recurrent modulation in V1 being associated with better (faster) target discrimination. To address this possibility, we compared the recurrent response in V1 at 250 ms as a function of response time (RT, fast, slow) based on a median split of RT on correct responses. Response time was on average 660 ms (SD 44 ms), the proportion of correct responses was 90.53% (SD 4.32%). The two target colors (red, green) did not differ with respect to response time and accuracy (all *P* values >0.1). The median split of RT was performed in each subject and experimental condition to compute individual trial averages (tERL, dERL) for fast and slow RTs. Those were then averaged separately as fast and slow trials over subjects to create the grand average waveforms and the CDR map in Fig. 4b. On fast trials, subjects responded on average 100 ms (SD 14 ms) faster than on slow trials.

Figure 4b plots the Nt-wave for fast (solid) and slow trials (dashed) measured at sensor sites representing the occipital–polar field effect (see inset) generated in V1. The CDR map shows the source distribution maximum of the fast-minus-slow difference at 250 ms, which confirms that the underlying source activity originates in V1. The response amplitude of the fast recurrent modulation in V1 is significantly increased for faster versus slower responses, consistent with an enhanced representation of the target at this level of representation. A further way to analyze the amplitude-performance link is to test whether the amplitude modulation of the V1 effect would systematically vary with the RT difference between fast and slow responses. Figure 4c plots the mean amplitude difference between slow and fast trials against the mean RT difference slow-minus-fast for each observer. The plot reveals that larger amplitude differences correlate with larger RT differences ($r = 0.396$, $P = 0.036$), indicating that the recurrent V1 effect is, in fact, linked to search performance with stronger recurrent activity in V1 leading to faster (correct) responses.

**ERMF topography and source localization of the Pd.** Figure 5 shows the magnetic waveform of the dERL (black) at selected sensor sites (see Methods). The Pd is highlighted as a green area under the curve between the ~200 and 300 ms. The field maps

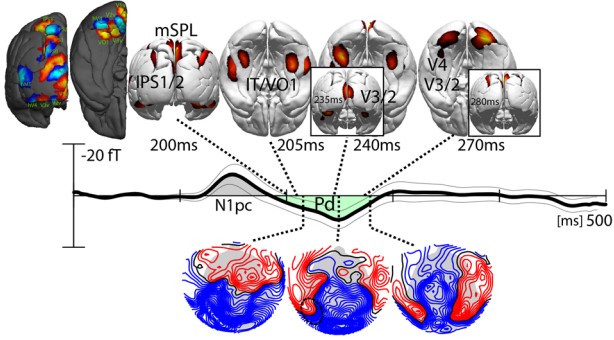

**Fig. 5 Event-related magnetic field (ERMF) distributions and current density reconstructions (CDR) of the Pd component.** The CDR maps show activity above an arbitrary threshold to illustrate the source maxima at representative time points after stimulus onset. The waveform in the replots the hemisphere-collapsed dERL response shown in Fig. 2. The ERMF maps show successive field distribution in the Pd time range. The field lines (blue/red) separated by increments of 2fT. The hot-scale and blue-scale areas rendered onto the dark gray hemispheres display probabilistic maps of retinotopic areas (see "Methods").

show the distribution of the Pd at time points best illustrating the field change over time. In comparison to the Nt (Fig. 3), the field polarity of the Pd (cf. dashed black circle) is opposite, with the lateral-temporal and medial-parietal field components representing the efflux and influx components, respectively. Similar to the Nt, the Pd starts as a modulation in higher-level extrastriate areas in anterior–lateral–ventral (IT/VO1), as well as in the dorsal parietal cortex (IPS1, IPS2). In contrast to the parietal sources of the Nt, which are left lateralized, the sources of the Pd are more balanced between the hemispheres. In addition, there is source activity in medial parts of the superior parietal lobe (mSPL), which is not seen in the Nt. The field distribution and CDR maxima of the ventral extrastriate sources then gradually propagate from more anterior–ventral to more posterior-occipital regions. (A movie showing the backward propagation of the Pd sources and field distribution is provided in Supplementary Movie 2). The ventral extrastriate sweep of backpropagation reaches early visual areas V2/V3 around 260 ms, which is ~20 ms earlier than the Nt sweep. Finally, around 280 ms, source activity reappears in parietal cortex areas that were active around 200 ms.

Importantly, similar to the occipital–polar field maximum of the Nt around 250 ms, there is a magnetic flux gradient over the occipital-central region between 200 and 240 ms, generated by a transient source maximum in early visual cortex (cf. inset), which ceases before the recurrent ventral extrastriate sweep of source activity reaches areas V3/V2 at ~270 ms. The occipital-central effect is of opposite polarity as compared to the Nt. In contrast to the Nt, however, the central occipital source maximum of the Pd does not originate in V1, but more dorsally, consistent with the lower VF representation of V2/V3. To assess whether the recurrent modulation maximum would link with search performance as it had for the Nt, we analyzed the response as a function of response time (fast, slow). The occipital-central waveforms show a slightly larger positive field response for fast versus slow trials between 200 and 250 ms after stimulus onset, consistent with a stronger opposite polarity (presumably inhibitory) modulation. An analysis of the amplitude variation in each subject as a function of the RT difference between the mean fast and slow responses, however, yielded no significant correlation ($r = -0.25$, $P = 0.11$). This implies that although the general activity pattern is similar between these two components, the functional role of the activity is different, with the fast back-propagating activity in V1 for the Nt resulting in a performance modulation, unlike the fast

back-propagating activity for the Pd, which neither reached that early area nor influenced performance.

**Comparing the source distribution of the Nt and Pd.** A comparison of the ERMF distribution and current sources of the Nt and Pd yielded striking similarities in terms of both the cortical localization and the temporal dynamics of the underlying activity. Both components reflect back-propagating activity in the visual cortex hierarchy starting in higher-level ventral (IT/VO) and dorsal stream areas (IPS0/1) and reaching early visual areas within 50–80 ms. Figure 6a shows the degree of overlap of the CDR distribution of the Nt (red) and the Pd (turquoise) between 180 and 300 ms. The maps display time-cumulative (collapsed over time) distributions arbitrarily thresholded to 40% of the CDR maximum in this time range. The overlap is considerable in mid- and lower-level extrastriate areas of the slow back-propagating sweep, which is consistent with the notion that both index a modulation process in the same cortical substrate with opposite polarity[18]. There are, however, topographical differences: (1) The fast recurrent sweep shows a maximum in V1 for the Nt, while the fast recurrent activity of the Pd locates to the lower VF representation of V2/V3. (2) The initial parietal activity of the Pd is more extensive and balanced between hemispheres. It locates to dorsal IPS areas (IPS1/2) and to an additional medial region of the SPL. The Nt, in contrast, shows left-lateralized parietal activity in IPS0/1/2. Furthermore, parietal source activity reappears around 280 ms for the Pd but not for the Nt. (3) The initial activity in higher-level ventral extrastriate areas is more dorsal for the Nt and more ventral and anterior for the Pd. To validate the topographical differences between the Nt and the Pd a statistical non-parametric mapping (SnPM[25], see "Methods" for details) analysis with the factors VF (left/right) and component (Nt, Pd) was performed for both the normalized ERMFs and the CDRs (MNLS). Figure 6b shows the VF (left/right) × component (Nt/Pd) interaction, which yields cortical regions where the distribution of the Nt and Pd differ significantly. The dashed trace plots log P values of the SnPM on the ERMFs between 0 and 500 ms, which yields three significant time ranges highlighted in gray. Topographical F-maps of SnPMs on the CDRs are shown at a representative time point in these ranges. The results validate the three major topographical differences described above. Around 190 ms significant differences appear in the parietal cortex due to source activity underlying the Pd being more extensive, with an additional medial superior parietal contribution not present in the Nt. Between 200 and 220 ms, there is a clear left lateral-anterior difference reflecting the more dorsal anterior–lateral sources of the Nt. The most significant source difference arises around 250 ms and locates to the primary visual cortex, reflecting the observation that the fast recurrent sweep of activation appears in V1 for the Nt, but not for the Pd. Finally, there is also a significant difference over the right parietal cortex at 260 ms, reflecting a reappearing parietal source activity (right hemisphere maximum) for the Pd but not for the Nt in this later time range (cf. Supplementary Movies 1 and 2).

## Discussion

One key observation we made here is that between 200 and 300 ms after stimulus onset source activity underlying the Nt and the Pd propagates from higher to lower hierarchical levels in the extrastriate visual cortex, suggesting that attentional processes of target enhancement and distractor attenuation operate in parallel as hierarchically recurrent sweeps of processing. This observation adds to previous documentations of attention-related back-propagation of activity in the ventral extrastriate cortex in humans[8,31,32] and in the monkey[33,34]. Importantly, we find that

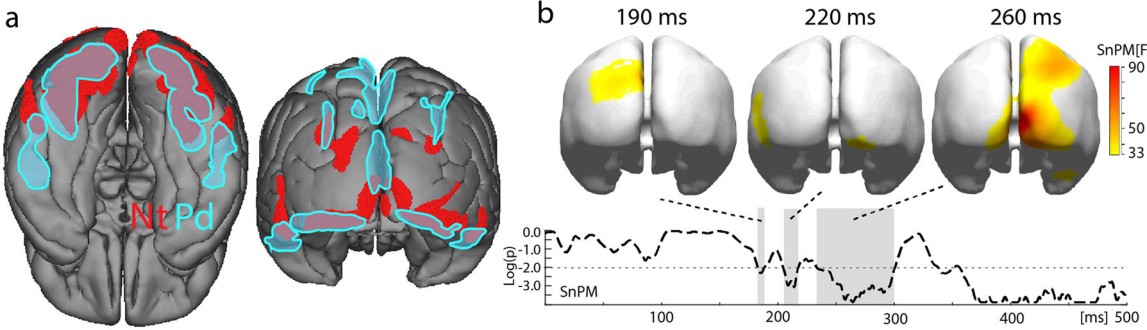

**Fig. 6 Overlap analysis of current source distribution underlying the Nt and Pd. a** Time-cumulative source distributions (180–300 ms) of the Nt (red) and Pd (turquoise). **b** Results of a statistical non-parametric mapping (SnPM) analysis comparing the Nt and Pd. Significant time ranges of topographical differences are highlighted in gray. The 3D-maps show the distribution of significant topographical differences at selected time points.

for the target-related modulation (Nt), this backpropagation involves a transient source maximum in primary visual cortex appearing prior to delayed activity in early visual areas V2/V3. Apparently, such temporal order of source activity does not fit a single back-propagating sweep of processing in ventral stream cortical areas that orderly reverses the cortical hierarchy level by level (assuming a cortical hierarchy as suggested by Essen et al.[35] and Felleman et al.[36]). Instead, the pattern suggests concurrent sweeps of feedback processing. One possibility is that a fast sweep originating in the IT cortex directly projects to the primary visual cortex and thereby overtakes a slower level-to-level back-propagating sweep in the ventral extrastriate cortex. The fast sweep would reflect activity bypassing intermediate levels of representation. Direct retrograde projections from high-level and mid-level ventral extrastriate areas to V1 have been documented[37–41]. Alternatively, the fast recurrent activation of V1 could reflect rapid feedback from dorsal stream areas starting in the parietal cortex[28,41–43] reaching V1 earlier than the slower recurrent ventral stream process starting in IT. Indeed, the Nt shows an initial source activity maximum around 200 ms in the parietal cortex (Fig. 3), which could be the source of this faster top-down modulation. With the present data, we cannot directly identify the cortical origin of the feedback modulation in V1. However, this observation adds to a range of experimental data suggesting that feedback projections to early visual cortex areas including V1 play a pivotal role in the attentional selection and visual awareness[27,29,33,44–50]. Recurrent processing is considered to be an essential mechanism in computational models to account for selective signal routing in the visual cortex[51,52] and to reduce the complexity of visual coding to render visual search a tractable problem[53,54]. If the fast recurrent process to V1 is critical for efficient search, it would be expected to vary with search performance. As shown in Fig. 4b/c, this is indeed the case. Stronger recurrent activity in V1 was associated with faster correct responses. Furthermore, for the target-related response (Nt) larger RT differences between fast and slow responses were mirrored by bigger amplitude differences of the recurrent modulation in V1, suggesting a direct link between recurrent activity in V1 and performance success.

What, then, is the functional role of the two concurrent sweeps of recurrent processing? The primary visual cortex has been proposed to serve as an active blackboard, with feedback to it being important for figure-ground segmentation, object-based attention, and working memory[27,30,55–57]. Information may be retroinjected into V1 from higher-level areas (parietal cortex), to refresh relevant parts of the input representation, to refine (increase the resolution of) an initial coarse representation provided with the initial feedforward sweep of processing[58,59], or to store intermediate computations in higher-level areas (active

cognitive blackboard)[30]. Observations with TMS[60] seem to support such an interpretation. Inactivation of V1 during visual search revealed two latency ranges after search frame onset during which discrimination performance was particularly disrupted, one around 100 ms, and a later phase between 200 and 400 ms. It was speculated that the early range reflects a repeated sampling of the input representation to extract the signal from noise, while the later phase reflects return projections from higher-level areas to V1. The here-observed concurrent fast- and slow-feedback sweeps would fit with this timing of recurrent processing and be compatible with an active blackboard model. The fast recurrent sweep to V1 would highlight the input representation of the search target or back-projects a coarse (interims) representation of the relevant item, while the simultaneous, but slower, a level-to-level process in ventral extrastriate cortex gradually tunes the resolution of discrimination at progressively lower levels of cortical representation to the spatial scale required for target discrimination[7,8,53]. When reaching the input level, the slow recurrent sweep will meet an already top-down biased representation of the target, which expedites its selection. A rapid top-down bias of the target representation (location) would also be consistent with the present observation that the Nt, but not the Pd, shows a recurrent sweep to V1, suggesting that the target but not the distractor is highlighted in the input representation. One may speculate whether the process reflects the realization of an enhanced saliency map in the primary visual cortex that guides the slow recurrent sweep to the target location.

Of note, it seems worth discussing the observation of concurrent fast and slow feedback in the framework of the predictive coding account[61–63]. A core notion of the account is that feedback from higher to lower-level visual areas carries predictions about activity changes in lower-level areas[62]. That is, the fast recurrent sweep may propagate predictions (built during the initial feedforward sweep of processing in the N1pc time range) about the target item (e.g., its location, orientation) to V1[64,65]. According to predictive coding, recurrent activity in V1 is largest when predictions are disconfirmed (residual error). We observe that when target selection is fast, the feedback response in V1 is biggest, suggesting that a larger residual error causes faster responses. At first glance, this does not reconcile with predictive coding, as a bigger residual error would be expected to involve additional processing, which postpones target discrimination. However, it is also possible that it is the error signal in V1 that provides the information critical for target discrimination, with a bigger signal eventually facilitating responses. Of course, with the present data, the nature of the recurrent modulation in V1 cannot be clarified. While the observed recurrent activity modulations reflect the operation of attention in the early visual cortex,

underlying information processing and cortical computations subject to those modulations are not easily inferable and must remain speculative.

Finally, except for color, subjects are unlikely to generate strong predictions about the target's orientation and location. The amplitude variation of the fast feedback-driven response in V1 may not reflect a prediction error. It may rather reflect ongoing intrinsic fluctuations of attention and/or the motivation to attend. Those may appear in form of more or less strong lapses or drifts of attention, which were in fact shown to be associated with attenuated activity in visual areas[66], consistent with the present observations.

Is the Pd a polarity reversed Nt? A notable feature of the Pd is that its topographical field distribution is similar to the Nt, suggesting that both components index a modulation process in the same cortical substrate, just with opposite polarity. The Nt may enhance, and the Pd may attenuate, the neural representation of the target and distractors, respectively[18]. Combined recording of FEF activity and posterior ERPs in the monkey revealed that target selection processes in FEF are the likely source of attentional modulation in the extrastriate cortex as indexed by the mN2pc[11,67]. The same FEF neurons showing a firing enhancement for target selection were found to display firing suppression for a salient distractor which was associated with a Pd-like ERP modulation in the monkey[12], consistent with the Nt and Pd response reflecting mirror image excitatory and inhibitory modulations in the same visual cortex areas. On the other hand, a topographical analysis of the Nt and Pd (spline-interpolation field maps) revealed that the Nt appears over more ventrolateral and the Pd over more dorsomedial cortex[15], suggesting that these components may not directly reflect modulations of activity in the very same cortical areas. The present topographical analysis of the current sources underlying the Nt and Pd reveals that both components are to large parts generated in the same regions of the ventral extrastriate cortex, but that there are also substantial differences. One is that the parietal activity underlying the Pd is more pronounced and widespread than that of the Nt. Notably, Hickey et al.[15] suggested based on the more dorsal topography of the Pd that it may represent the parietal subcomponent of the N2pc identified with MEG recordings[13]. The present data partially endorse this interpretation, insofar as parietal source contributions to the Pd are stronger than to the Nt. Nonetheless, substantial portions of source activity of the Pd are generated in ventral extrastriate areas and overlap with sources of the Nt, which rules out that the Pd is a parietal response only.

A further difference between the Nt and the Pd is that the former shows stronger source activity in the left than the right ventral extrastriate areas—an asymmetry for which we have no clear explanation. Left-lateralized N2pc responses have been described previously when searching for items requiring a lexical discrimination[3,68]. The present experiment used characters (T), but the task was to discriminate the color and orientation of the characters which would not involve a lexical decision. Other research, has suggested that the categorical perception of the target colors is associated with a left-lateralized N2pc in visual search[69]. The present experiment did not require categorical discrimination of the target color. However, it is possible that the target and distractor color activated competing for categorical representations of color, which could have led to a stronger response in the linguistically specialized left hemisphere. Based on the present data we can only speculate, but if this interpretation applies, it is notable that the Pd does not show such hemisphere lateralization.

**Conclusion.** The present work provides a detailed spatiotemporal analysis of source activity generating the Nt and Pd components

of the N2pc. We find that source activity underlying both components propagates from higher to lower hierarchical levels in the visual cortical hierarchy, suggesting that cortical processes of target enhancement and distractor attenuation operate in parallel and in the reverse hierarchical direction in the visual cortex. Most importantly, source activity underlying both components consists of parallel-operating fast and slow back-propagating sweeps, with the fast sweep of the Nt, but not the Pd, directly reaching the primary visual cortex within ~50 ms after activity onset in higher-level areas, but ~25 ms prior to the slower sweep passing through progressively lower levels of representation (V4-V3-V2) in ventral extrastriate cortex. These observations emphasize the seminal role of recurrent processing for both target selection and distractor attenuation. More importantly, they reveal that recurrent processing is mediated by multiple fast and slow sweeps of back-propagating activity, with the fast sweep being critical for successful target discrimination presumably by enhancing the representation of relevant input in the primary visual cortex.

## Methods

The data reported in the current study have been previously reported (with the focus of the analysis being different) as Experiment 1 in Donohue et al.[23].

**Participants.** Twenty-four human participants were included in this study (10 male, ages: 22–34, mean = 25.4 years old, 1 left-handed). Additionally, five subjects were excluded because 30% or more of their trials contained physiological artifacts (e.g., blinks, horizontal eye movements). Written, informed consent was obtained for all participants and all methods and procedures were approved by the ethics committee of the Otto-von-Guericke University, Magdeburg.

**Stimuli and task.** The visual search task consisted of an array of six "T"s, all equidistant in a circle, 7.4° from the central fixation cross (see Fig. 1). One of the "T"s was always green, another was always red, and the rest of the "T"s were always blue, all presented on a gray background. The pop-out "T"s (red and green) were equated for luminance. The size of each T took up 3.4° of visual angle. The search array was presented for 300 ms, after which the fixation cross remained on the screen for 1100–1600 ms (jittered) before the next trial began.

The task of the participant was to, in a given block, find the T in the target color (e.g., red) and determine if it was tilted to the left or to the right via button press. For half of the blocks, participants attended to the red T, thereby making the green T the pop-out distractor, and for the other half of the blocks, participants attended to the green T, making red the pop-out distractor. The target color switched blockwise between red and green, and participants were randomly assigned one color as the starting color for the first block. Participants' responses were recorded and counted as correct if they occurred between 200 and 1000 ms post array onset. Prior to the start of the experiment, participants were given a practice block to ensure they were familiar with the task, and their fixation and head position in MEG were monitored online, with feedback given at the end of each block.

The location of the respective target and distractor Ts was randomly assigned trial-to-trial with a few limitations. In 40% of the trials (n = 960), the target was presented on the vertical meridian and the distractor was presented in one of the lateral positions. In 40% of the trials (n = 960), the distractor was presented on the vertical meridian and the distractor was presented laterally. Finally, in 20% of the trials (n = 480), both the target and the distractor were presented laterally (always on opposite sides of the display). For the present data, the trials in which both the target and distractor were lateralized were not analyzed.

**MEG acquisition and analysis.** MEG data (along with simultaneous EEG/VEOG data, not reported here) were recorded continuously with 248 magnetometers from a BTI Magnes 3600 whole-head system at a sampling rate of 508 Hz, with the low-pass filter of the system set to DC to 50 Hz. To eliminate the influence of environmental noise, the BTI system has reference coils, which were used to cancel such environmental noise online[70].

Offline, epochs spanning from 200 ms before and 500 after stimulus onset were extracted from the raw data. These epochs were subjected to artifact rejection using MSI software with the thresholds for rejection separately determined for each subject. Across participants, an average of 14% of trials (n = 336 of the total of 2400 trials) was rejected due to eye blinks, eye movements, and other physiological noise. The event-related magnetic field response was derived by computing selective data averages for epochs with the same target and distractor positions, thereby collapsing across the two possible target colors. Specifically, for each subject and sensor position, four average ERMF waveforms were computed for epochs (1) where the target appeared in the left VF with the distractor appearing at the upper or lower vertical meridian position, (2) where the target appeared in the right VF with the distractor appearing at the upper or lower vertical meridian position, (3) where the

distractor appeared in the left VF with the target presented at the upper or lower vertical meridian position, and (4) where the distractor appeared in the right VF with the target presented at the upper or lower vertical meridian position.

**Data co-registration and re-positioning**. In each subject, the MEG sensor positions were co-registered with individual anatomical landmarks (nasion, left and right preauricular point) as well as five localizer coils placed at standardized positions in an EEG cap (Easycap, Herrsching, Germany) using the Polhemus 3Space Fastrak system (Polhemus, Colchester, VT, USA). Because each participant's head was located in a slightly different position within the MEG sensor array, the data were repositioned offline to a reference sensor array (grand average over 1500 recording sessions) before computing the grand average. To this end, the individual sensor space data were transformed into a source space representation (minimum norm least-squares estimates) using a canonical lead field (MNI brain). The data were then back-projected to the reference sensor space via lead-field inversion.

**Current source localization**. Current density reconstructions (CDRs) shown in Figs. 3, 4, 5, and Supplementary Fig. 1 were computed based on the sensor-level (all 248 magnetometers) grand average ERMF responses over subjects. CDRs were estimated with the minimum least-squares (MNLS) method as implemented in the multimodal neuroimaging software Curry 8 (Compumedics USA, Inc.). Mathematical details of implementation in Curry 8 are described in refs. [71,72]. To provide a brief description, the source localization analysis here used a distributed source model (source density model) in which a lead field (forward model) $(m_f = Lj)$ links the current source vector (fixed-location and fixed-orientation dipoles) with the magnetic field vector (sensor data), with $m_f$ being the forward calculated data vector, L the lead-field matrix, and j the vector containing the dipole components. Source estimates were computed by minimizing (1) $\Delta^2 = D(j) + \lambda M(j)$, with $D(j)$ and $M(j)$ representing the data and the model term, respectively. $\lambda$, the regularization parameter, balances the goodness of fit and the closeness to the model. The MNLS approach minimizes (2) $D(j) = ||Lj - m||^2$ (m is measured data), and $M(j) = || Wj||^2$, with W representing a diagonal location weighting matrix. The latter serves to compensate for the undesired depth dependency of the standard MNLS approach. The optimal value for the regularization parameter was determined by using the $\chi^2$ criterion based on the assumption that $\Delta^2$ should be roughly in the range of the noise in the measured data[72]. The noise was estimated relative to the 200 ms baseline ERMF response immediately prior to search frame onset. Dipole locations and orientations of the cortical source compartment were defined using a high-resolution 1-mm-triangulation (BEM)[73] of the MNI152 brain (gray-to-white matter border) which yielded $n = 287{,}481$ current dipole locations (nodes of the triangularization) with a fixed surface-normal orientation. The cortical surface segmentation was performed with FreeSurfer (version 5.1.) and FSL (http://www.fmrib.ox.ac.uk/fsl/) and then transformed and imported into Curry 8 using a custom made Matlab script (https://doi.org/10.17605/OSF.IO/DYUG6). Respective segmentation, used to define the source compartment of the current density reconstructions, also served as the cortical surface on which the probabilistic maps of retinotopic visual areas (shown in Figs. 2 and 3) were rendered (see description below). In Curry 8, the BEM was constructed using the image analysis module.

It should be noted that the accuracy of source localization based on grand average data is limited because subjects vary in their exact geometry of cortical gyri and sulci. Averaging the ERMF response across subjects will therefore smooth over individual differences. We assume that the cortex structure of the MNI152 brain provides a representative source compartment, as cortical smoothing due to anatomical averaging will approximately match the ERMF smoothing due to data averaging.

For the statistical comparison of source activity underlying the Pd and Nt (SnPM described below), CDRs were computed in each individual subject using the MNLS approach described above. A source compartment with a lower resolution segmentation of the MNI brain (standard brain, provided in Curry 8) and a smaller number of dipoles ($n = 9627$) was used. This was necessary due to the technical limitations (computer memory), as the computations to derive the SnPMs could not be performed using high-resolution estimates of cortical activity. Beforehand the sensor positions of each subject were brought into the common register by realigning the landmarks and marker coil positions to a reference sensor array, as outlined above.

**Statistics and reproducibility**. Time ranges of significant ERMF variation were determined using a topographic analysis of variance (TANOVA)[24,25] based on non-parametric permutation as implemented in the multimodal neuroimaging software Curry 8. The analysis was performed for subsequent time samples, with significant samples defining the range of significant ERMF variation. Significance was defined as independent of the underlying topographical distribution of the magnetic field response. The tERL and dERL responses were separately analyzed using a one-way TANOVA with the factor visual field of target/distractor presentation (left, right). The computation was performed time sample-by-sample by considering the response at all sensors in a time range between 0 and 500 ms after stimulus onset. Control for multiple comparisons in the time domain was based on spectral properties (sampling frequency ($f_s$) and the cutoff frequency ($f_c$))

of the ERMF signal, with the corrected alpha-level being: (3) $\alpha\text{-corr} = 1 - (1 - \alpha\text{-nom})^{2fc/fs}$ ($\alpha$-corr—corrected alpha-level; $\alpha$-nom—nominal alpha-level (0.05)). For details and the reasoning behind the derivation, see Wagner et al.[74]. All reported statistical parameters represent corrected values. Significant effects were only considered in time ranges where each event type (left and right VF responses) passed the topographical consistency test (TCT)[24], which is a randomization test (based on shuffling data among sensors) of the consistency of a topographical pattern reappearing in each observation (trial) of a given experimental condition (MATLAB code of TCT is provided in Koenig and Melie-García[24]). TCT, therefore, limits statistical comparisons between experimental conditions to time ranges in which the Nt and Pd are consistently present in the ERMF distributions. For downstream statistical analyses, only time ranges of significant topographical ERMF effects were considered.

The statistical comparison of source activity between the Nt and Pd was performed using SnPM as implemented in Curry 8[25,74]. SnPM is based on non-parametric permutation testing and involves control for the family-wise error rate (dipole locations). It allows us to test whether current density distributions (or sensor data) differ in a significant manner. SnPM was computed based on CDR maps of each subject and experimental condition and performed for successive time samples between 0 and 500 ms. SnPM involved consistency testing (TCT) within conditions. Hence, significant Nt × Pd SnPMs were considered only in time ranges with consistent source activity for each event type across subjects. As for the TANOVA outlined above, control for multiple comparisons in the time domain was based on spectral properties of the ERMF data as suggested by Wagner et al.[74].

For statistical validation of the correlation (Pearson's linear r) between the Nt/Pd amplitude differences and the response time difference of fast and slow responses, random permutation testing[75] ($n = 1000$ permutations) at a family-wise alpha-level of 0.05 was used.

The standard error (SE) of the ERL difference waves (target/distractor contralateral minus ipsilateral to sensor hemisphere) shown in Figs. 2, 3, and 5 (thin gray traces) represents the standard error of the difference between means: (4) $SE = \text{sqrt}(\sigma_{contra}^2/n + \sigma_{ipsi}^2/n)$, with $\sigma$ being the standard deviation and n the number of subjects.

**Probabilistic maps of retinotopic visual areas**. The retinotopic areas shown in Figs. 2 and 3 represent probabilistic maps available at https://scholar.princeton.edu/napl/resources[76,77], which were rendered onto a 1-mm surface segmentation of the MNI152 brain. For visualization purposes, the spatial extension threshold of each of the shown areas (V1, V2, V3, hV4, hMT, VO1, IPS0-3) was arbitrarily set to avoid strong overlap among neighboring areas.

**Reporting summary**. Further information on research design is available in the Nature Research Reporting Summary linked to this article.

## Data availability
The datasets generated during and/or analyzed during this study are available from the corresponding author on reasonable request. All source data underlying the graphs and charts presented in the main figures are provided in Supplementary Data 1.

## Code availability
Custom code used for data analysis and transformation is available from the corresponding author on request as well as download from OSF (https://doi.org/10.17605/OSF.IO/DYUG6)[78].

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

## Acknowledgements
This work was made possible by DFG Grants SFB 779, TP-A1, TP-A14N, DFG Grant Ho-1965-2/1.

## Author contributions
S.E.D., M.A.S., and J.-M.H. designed research; S.E.D. performed research; S.E.D. and J.-M.H. analyzed the data; S.E.D. and J.-M.H. wrote the paper.

## Funding

## Competing interests
The authors declare no competing interests.
