## [Peer Review File · Communications Biology]

Reviewers' comments:

Reviewer #1 (Remarks to the Author):

The authors use magnetoencephalography to investigate the neural substrate of mechanisms involved in boosting target representations and suppressing distractor representations during attentional selection and resolution.

The results suggest that attentional selection causes a quick reentrant feedback loop from higher visual areas to V1 and other lower visual areas. Something similar occurs during distractor suppression, with quick feedback from high to low areas impacting the representation of unattended stimuli. However, while reentrance on target representations begins very early in the visual hierarchy (V1), distractor suppression acts on later stage representations (V2/V3). This result importantly and novelly differentiates the nature of mechanisms involved in enhancement and suppression.

The paper does:

- provide a level of detailed insight into attentional mechanisms that is uncommon in the literature.
- provide solid support for broadly accepted and influential ideas about how selective attention operates.
- constitute one of the first MEG investigations of distractor suppression, showing that the electrical signature of suppression emerges in magnetic recordings and is robust enough to be employed in experimental work.
- draw relationships between observed physiological effects and overt behaviour.

The paper does not:

- Test contentious theoretical issues. The experiment is not designed to differentiate between current theoretical perspectives on selective attention.

I quite like the paper and I expect that it will be highly cited.

I have very little in the way of substantive criticism. That is, the experimental design is tried and true, and is suitable for the purpose of eliciting attentional mechanisms acting on target and distractor representations. The analytic approach is clear and well-justified (there is room for more detail on statistical approaches; more on this below). Techniques like source-analysis use conservative and well-verified approaches, making it easy for the reader to trust the results. The results are carefully interpreted and explained and the paper is very well written.

Minor comments:

'There is now substantial experimental evidence...' I miss a reference here to our paper looking at the relationship between Pd and eye movement indices of distractor suppression (Weaver, van Zoest, & Hickey, 2017). On the other hand, the paper from Gaspelin, Leonard, & Luck (2015) cited here is an analysis of eye movements, does not involve ERPs.

'The N1pc will not be in the focus here.' Slightly awkward, consider active voice 'We will not consider the N1pc further in this paper.' You might direct the reader to the earlier paper in this context.

'To verify time ranges of significant topographical variation of the tERL and dERL...' Some further discussion of this method might be in order. This gives impression that this statistical result here somehow related to the emergence of specific topographies (eg. microstates). But I don't think this is the case.

'Also, the polarities of the magnetic waveform differences reported here are arbitrary...' May be worth clarifying here that while the directionality is arbitrary, the opposition of polarity in Nt and Pd emerges in the magnetic field as it does in the voltage potential (as is made clear elsewhere in the manuscript).

'The plot reveals that larger amplitude differences correlation with larger RT...' Pearson product moment is potentially unrepresentative with low sample size and outlier values. Do results emerge in Spearman's rho? Alternatively, is the Pearson result significant when significance is tested using permutation test (which implicitly takes outlier values into consideration)?

'Additionally, five subjects were excluded because of physiological artifacts.' On what criteria were these participants excluded?

'Predictive coding predicts, however, that recurrent activity in V1 is largest when predictions are disconfirmed.' I'm not sure the results are at odds with this, as the design doesn't generate strong a priori expectations prior to stimulus presentation. If participants were to expect a T, but get a rotated T, then I do expect that you'd see the increase in reentrance mentioned here. The fact that 'fast feedback response in V1 is biggest when target selection is fast' is probably related to motivation, and doesn't seem inconsistent with predictive coding. That is, when participants are motivated, they front-load the operations required to make a response and pack these operations early in the cognitive process. This created a 'bigger' fast feedback response (because what is otherwise spread out over time occurs in a shorter interval) and the size of this early response predicts RT. In any case, this passage seemed underdeveloped, worth further consideration.

signed, Clayton Hickey

Reviewer #2 (Remarks to the Author):

This study has mapped the cortical localization of event-related activity recorded with current EEG and MEG for visual search task. It was found that, N2 component of the evoked activity responding to target selection, has distributed sources along the visual cortical hierarchy and that the sequence of activation of different cortical regions is suggestive of the hierarchical processing. The study is well performed, and the analysis seem at most part sounds. My major concern is that numerous methodological details are missing and hence the correctness of the approach is impossible to evaluate. This is especially true for the source-reconstruction of MEG-EEG data albeit it was the main approach in the study. Furthermore, albeit the results support the presence of wide-spread cortical sources underlying the generation of N2 component, this does not necessarily indicate hierarchical processing and thus conclusion of the study are slightly over interpreted.

Major

- There was no description in the method section, how event related activity was estimated from and MEG data. I understand that these procedures are rather standard, but they nevertheless, should be described in the Methods.

- The main aim of the study was to estimate cortical sources underlying the generation of N2 component for target detection in visual search task. To this end, source reconstruction of MEG-EEG data was performed with current source density (CSD) analysis. Numerous methodological details were missing from the description of source reconstruction. Was the source modelling

approach performed separately for each subject? Was it performed for "raw" ME-EEG time-series data i.e were cross-spectral density (CSD) matrices computed as a function of time for all channel pairs or were the event-related components estimated from sensor-level EEG or MEG data and then modelled? If the latter was the case, please describe how this was done, how were the time windows and channels selected? Was the source modelling performed for each trial or were the trials first averaged across the sessions (which trials) and then modelled. It would be very good, if the formulas were given to indicate exactly how the source reconstruction was performed and how data were averaged (See Vliet et al., 2018 *Frontiers in Neuroscience*).

- Was minimum norm least-squares algorithm used to estimate cortical sources from CSD matrixes? Overall, there is no description of how forward and inverse models were created. Please add.
- It is written that "CSDs of each individual subject were computed using a lower resolution segmentation of the MNI brain". What does this sentence mean? Please clarify in which resolutions were the computations carried out.
- How was the source modelled data averaged across subjects and used in the statistical analysis?
- The statistical comparison of source activity between the Nt and Pd was performed using statistical non-parametric mapping (SnPM). Please give more details of this analysis.
- The source activity underlying the generation of N2 is estimated from several time points. As the cortical activity is seen to shift across time, this is interpreted as an indication of hierarchical processing of search information, which is currently the main conclusion in the abstract. This is overinterpretation of the results as there is no analysis of the direction of the information flow and the results only show that the strongest cortical sources vary in time.
- Confidence limits should be added to the waveforms in Fig. 2,4, 4A, 5 and 6.
- Figure 3, 5 and 6 shows topographies and source locations above arbitrary threshold. Why is this threshold arbitrary and not a statistical threshold? These figures display the main results of the study.
- The result section is a list of sequence of cortical activations and it tedious to read. It would benefit from editing and adding some explanation why these results are important.

Minor

- How many trials remained after artefact removal and were used for the analyses?
- Averaged brain and not individual MRIs were used for source reconstruction. Since the functional anatomy varies rather largely across subjects, this approach does not yield very accurate source localizations. It should be discussed how this impacts the localization of the cortical sources of the N2 components.
- What was the band-pass filtering for the collection of MEG-EEG data? There are always a low- and high pass filters in the recording hardware.
- Was ICA used to remove artifacts caused by eye blinks and heart beats?

- The statistical analysis was carried out sample-by-sample and Correction for (temporal) multiple comparison was performed as in ref 69. Please give a brief description also in the present manuscript.
- There is no description of the behavioral performance in the Result section nor in the Methods. Albeit, data is also reported in Donohue et al., 2018, sufficient details should also be give here.

We would like to thank you for considering our work. We also thank the reviewers very much for their time and effort to provide helpful comments and suggestions. We believe we could address all issues that were raised in a satisfying manner. All changes are detailed below in a point-by-point response to the reviewers. Text changes are highlighted in red in the manuscript.

Reviewer #1

We are happy that reviewer #1 received our work with enthusiasm. He had only minor issues.

(1) 'There is now substantial experimental evidence...' I miss a reference here to our paper looking at the relationship between Pd and eye movement indices of distractor suppression (Weaver, van Zoest, & Hickey, 2017). On the other hand, the paper from Gaspelin, Leonard, & Luck (2015) cited here is an analysis of eye movements, does not involve ERPs.'

As suggested, we added the suggested citation (Weaver, van Zoest, & Hickey, 2017) and removed Gaspelin et al. (2015).

(2) 'The N1pc will not be in the focus here.' Slightly awkward, consider active voice 'We will not consider the N1pc further in this paper.' You might direct the reader to the earlier paper in this context.'

We corrected the 'awkward' wording of the sentence as suggested.

(3) 'To verify time ranges of significant topographical variation of the tERL and dERL...' Some further discussion of this method might be in order. This gives impression that this statistical result here somehow related to the emergence of specific topographies (eg. microstates). But I don't think this is the case.'

We agree that the wording here implies that the TANOVA aims at deriving some sort of topographical microstates, which was not intended. We use the TANOVA solely to derive time ranges in which the experimental conditions differ significantly, independent of the underlying topographical variation. We have extended the Methods section to emphasize this point. We also reworded the results section to eliminate the misleading focus on topographical differences.

(4) 'Also, the polarities of the magnetic waveform differences reported here are arbitrary...' May be worth clarifying here that while the directionality is arbitrary, the

opposition of polarity in Nt and Pd emerges in the magnetic field as it does in the voltage potential (as is made clear elsewhere in the manuscript).'

We added text clarifying the issue as suggested.

(5) 'The plot reveals that larger amplitude differences correlation with larger RT... Pearson product moment is potentially unrepresentative with low sample size and outlier values. Do results emerge in Spearman's rho? Alternatively, is the Pearson result significant when significance is tested using permutation test (which implicitly takes outlier values into consideration)?'

We thank the reviewer for pointing this out. We agree that parametric testing would be potentially unrepresentative. We therefore followed the suggestion and used permutation testing to verify the significance of Pearson's r . The analysis did not change the results and led only to a minimal change of the p-values. Relevant information about the permutation test (# of permutations, family-wise error rate) is added in the Method section.

(6) 'Additionally, five subjects were excluded because of physiological artifacts.' On what criteria were these participants excluded?'

Five additional subjects were excluded because they had more than 30% of their total trials rejected due to physiological artifacts (horizontal eye movements and blinks). We added the information to the Methods section.

(7) 'Predictive coding predicts, however, that recurrent activity in V1 is largest when predictions are disconfirmed.' I'm not sure the results are at odds with this, as the design doesn't generate strong a priori expectations prior to stimulus presentation. If participants were to expect a T, but get a rotated T, then I do expect that you'd see the increase in reentrance mentioned here. The fact that 'fast feedback response in V1 is biggest when target selection is fast' is probably related to motivation, and doesn't seem inconsistent with predictive coding. That is, when participants are motivated, they front-load the operations required to make a response and pack these operations early in the cognitive process. This created a 'bigger' fast feedback response (because what is otherwise spread out over time occurs in a shorter interval) and the size of this early response predicts RT. In any case, this passage seemed underdeveloped, worth further consideration.'

We agree with reviewer that the paragraph is somewhat underdeveloped, not least because the exact functional role of the V1 modulation cannot be clarified with the reported experiment and must remain, to some extent, speculative. We have extended the discussion to cover the issues raised by the reviewer. We put into perspective the conclusion that the V1 signal enhancement is incompatible with

predictive coding, by also entertaining the possibility that the V1 effect reflects the revision of the prediction error, which would then provide the information that facilitates target selection. Furthermore, we now conclude the paragraph by discussing the possibility that the V1 signal variation does not reflect a prediction error at all, but rather ongoing intrinsic fluctuations of attention (akin to lapses of attention) or fluctuations of the motivation to attend, as hypothesized by the reviewer.

Reviewer #2

Major issues

(1) *'There was no description in the method section, how event related activity was estimated from and MEG data. I understand that these procedures are rather standard, but they nevertheless, should be described in the Methods.'*

As suggested, we have added to the Methods section a more detailed description of how the event-related magnetic field responses were derived for each experimental condition (under MEG acquisition and analysis).

(2) *The main aim of the study was to estimate cortical sources underlying the generation of N2 component for target detection in visual search task. To this end, source reconstruction of MEG-EEG data was performed with current source density (CSD) analysis. Numerous methodological details were missing from the description of source reconstruction. Was the source modelling approach performed separately for each subject? Was it performed for "raw" ME-EEG time-series data i.e were cross-spectral density (CSD) matrices computed as a function of time for all channel pairs or were the event-related components estimated from sensor-level EEG or MEG data and then modelled? If the latter was the case, please describe how this was done, how were the time time-windows and channels selected? Was the source modelling performed for each trial or were the trials first averaged across the sessions (which trials) and then modelled. It would be very good, if the formulas were given to indicate exactly how the source reconstruction was performed and how data were averaged (See Vliet et al., 2018 Frontiers in Neuroscience).*

One major concern of the reviewer is that insufficient detail is provided in describing our approach to current source analysis. We have accordingly revised and extended the Methods section to provide respective information. Specifically, we clarify that source modelling was performed on raw-ERMF time-series data (sensor-level data) averaged over subjects. We did not reconstruct sources using trial-based estimates in individual subjects to subsequently average data in source space. We also did not analyze our data in the spectral domain, i.e. we did not compute cross-spectral density matrices. The reviewer, however, points us to the issue that the abbreviation

CSD for current source is somewhat unfortunate, as it overlaps with the term cross-spectral density. To avoid eventual misinterpretations, we have replaced CSD by the equivalent term CDR (current density reconstruction) throughout the manuscript. Finally, as suggested by the reviewer, we added the essential formula to outline the mathematical approach to CDR, as implemented in Curry 8.

(3) 'Was minimum norm least-squares algorithm used to estimate cortical sources from CSD matrixes? Overall, there is no description of how forward and inverse models were created. Please add.'

Yes the MNLS algorithm was used to derive CDRs, which is now indicated in the Methods section. We also added a description and formulas of the forward and inverse model in the Methods section.

(4) 'It is written that "CSDs of each individual subject were computed using a lower resolution segmentation of the MNI brain". What does this sentence mean? Please clarify in which resolutions were the computations carried out.'

We have clarified this point in the Methods section by annotating that the reduced resolution of the sources space was necessary due to technical limitations and computing power. Computing power (and the implementation of SnPM in Curry 8) was not enough to compute the SnPMs based on the high-resolution segmentation of the MNI152 cortex, which contained 287481 nodes (dipoles) per subject ($n=24$). A lower resolution segmentation with 9627 nodes, however, allowed us to perform the necessary computations. As requested, we provide the resolution of computations as number of dipoles on the MNI-surface in the Methods section.

(5) 'How was the source modelled data averaged across subjects and used in the statistical analysis?'

Except for the direct comparison of the Nt and Pd distribution, which used SnPM on individual CDRs (Figure 6), the statistical analysis of all other data was performed in sensor-space using the TANOVA approach. The CDRs, which were computed based on sensor-space averages over subjects, were not used in the TANOVA.

(6) 'The statistical comparison of source activity between the Nt and Pd was performed using statistical non-parametric mapping (SnPM). Please give more details of this analysis.'

As suggested, we have added more details in the Methods section describing details of the SnPM approach.

(7) 'The source activity underlying the generation of N2 is estimated from several time points. As the cortical activity is seen to shift across time, this is interpreted as an indication of hierarchical processing of search information, which is currently the main conclusion in the abstract. This is overinterpretation of the results as there is no analysis of the direction of the information flow and the results only show that the strongest cortical sources vary in time.'

We would like to emphasize that the conclusions about the hierarchical processing are not implied to show information flow. They exclusively refer to the observation that activity modulations due to attention propagate from higher to lower levels in the cortical processing hierarchy, which is best illustrated in the supplementary movies. We entirely agree with the reviewer that from the back-propagation of those modulations we cannot infer information flow. What we show is a modulation of neural activity only as indexed by changes of source activity. The eventual flow of search information from higher to lower hierarchical levels is not accessible with our experimental approach, and it is also not an intended conclusion to be drawn from the data. We changed the Abstract and added text in the discussion section (paragraph on predictive coding) to clarify this point.

(8) 'Confidence limits should be added to the waveforms in Fig. 2,4, 4A, 5 and 6.'

As suggested, we have added confidence intervals to Figures 2,3, and 5, which show average ERMF waveform differences. For Figures 4A and 6, confidence intervals could not be derived, because the source waves shown in 4A are taken from the CDR of the grand average ERMF response. The trace in Figure 6 shows p-values.

(9) 'Figure 3, 5 and 6 shows topographies and source locations above arbitrary threshold. Why is this threshold arbitrary and not a statistical threshold? These figures display the main results of the study.'

The presence of back-propagating Nt and Pd activity is statistically verified with the TANOVA analyses. The thresholds are, in fact, not completely arbitrary as they are set on each time point to a cut-off of 40% of the maximal source density in the map (this also applies to the supplementary movies). In that way, the thresholds of the CDR maps are set to best illustrate the propagation of source activity, which would lose illustrative power when thresholded with a statistical limit.

On a more general note, it is hard to define a meaningful significance threshold from an inverse source model of a single grand average ERMF data set. The only way to derive some sort of statistical criterion would be to set the threshold to say the upper 5 % of the total amplitude variation of source activity in the whole map. This would however not be very meaningful, because this criterion would yield significant sources even in a CDR map estimated from complete noise. As such, we

chose to use the 40% threshold, as we found it to best reflect the significant patterns of activity for display purposes.

(10) **'The result section is a list of sequence of cortical activations and it tedious to read. It would benefit from editing and adding some explanation why these results are important.'**

We have added to this section to make it more clear to follow and to add brief explanations of what results are important and what this implies (though more of this is, of course, left to the discussion).

Minor issues

(1) ***'How many trials remained after artefact removal and were used for the analyses?'***

We have added the number of rejected trials in the Methods section.

(2) ***'Averaged brain and not individual MRIs were used for source reconstruction. Since the functional anatomy varies rather largely across subjects, this approach does not yield very accurate source localizations. It should be discussed how this impacts the localization of the cortical sources of the N2 components.'***

We have added a paragraph to the Methods section discussing the data smoothing effect of using segmentations of an average brain for source reconstruction in individual subjects.

(3) ***'What was the band-pass filtering for the collection of MEG-EEG data? There are always a low- and high pass filters in the recording hardware.'***

We apologize that this was not clearly stated. We have now added text explicitly mentioning that BTI Magnes system was recording in DC-coupled mode with an inbuilt low-pass filter set to a 50 Hz cutoff.

(4) ***'Was ICA used to remove artifacts caused by eye blinks and heart beats?'***

No, we opted to use artifact rejection only, as it avoids the risk of eventually removing signal with ICA correction.

(5) ***'The statistical analysis was carried out sample-by-sample and Correction for (temporal) multiple comparison was performed as in ref 69. Please give a brief description also in the present manuscript.'***

We have added text and formula in the Methods section to explain the approach to, and the reasoning behind, the multiple comparisons correction in the time domain.

(6) *'There is no description of the behavioral performance in the Result section nor in the Methods. Albeit, data is also reported in Donohue et al., 2018, sufficient details should also be give here.'*

We added respective information to the Methods section (under Stimuli and task).

REVIEWERS' COMMENTS:

Reviewer #1 (Remarks to the Author):

The authors have addressed the concerns I raised in my earlier review. It's a very nice paper and I look forward to seeing it in print.

Reviewer #2 (Remarks to the Author):

The authors have revised the manuscript according to suggestion and added sufficient details of the methodological approaches to Method section. Few minor issues remains to be clarified.

1. ERMF distributions with current density reconstructions (CDR) show that EMRF activity is localized to various brain regions. Clarify how source vertices were mapped to or parcellated into these brain regions.
2. Figure 3 shows the field distribution corresponding with the CDR maps. Please clarify how the dipoles were averaged to yield a source waveform presented in the Figure.
3. Fig. 4B shows correlation of Nt with the RT and 4C the RT differences. RT distributions underlying these results are now reported in the Methods Section. I suggest moving these text to the results section before going into the results of Fig4B.
4. The division to fast and slow RTs used in the analysis in Fig. 4B should be clarified. In the result section, it is described that "To address this possibility, we compared the recurrent response in V1 at 250 ms as a function of response time (RT, fast, slow) based on a median split of RT (within subjects) on correct response". Does this mean that trials for each subject were divided to those with fast and slow RTs or that subjects were divided to fast and slow responders? Please report the median RTs that were used for splitting.

Response to reviewer #2

The authors have revised the manuscript according to suggestion and added sufficient details of the methodological approaches to Method section. Few minor issues remains to be clarified.

1. ERMF distributions with current density reconstructions (CDR) show that EMRF activity is localized to various brain regions. Clarify how source vertices were mapped to or parcellated into these brain regions.

We have added clarifying text in the Methods section explaining that the same segmentation (triangularization) of the MNI152-brain was used for rendering the brain regions (probabilistic maps) using Freesurfer and for the source density reconstructions created in Curry 8. This guaranteed that the ROIs in Figures 3 and 5 perfectly fit the source compartment of the CDR estimates.

2. Figure 3 shows the field distribution corresponding with the CDR maps. Please clarify how the dipoles were averaged to yield a source waveform presented in the Figure.

The waveform in Figure 3 is a replotting of the ERMF waveform shown in Figure 2. The waveform represents the magnetic field response (not a source wave) at the sensor sites highlighted in the inset in Figure 2. It is re-plotted in Figure 3 to give the reader orientation regarding the approximate temporal latency of the shown source activity distributions. We have added a sentence in the Legends of Figure 3 and Figure 5 to clarify this issue.

3. Fig. 4B shows correlation of Nt with the RT and 4C the RT differences. RT distributions underlying these results are now reported in the Methods Section. I suggest moving these text to the results section before going into the results of Fig4B.

We have moved the text as suggested.

4. The division to fast and slow RTs used in the analysis in Fig. 4B should be clarified. In the result section, it is described that "To address this possibility, we compared the recurrent response in V1 at 250 ms as a function of response time (RT, fast, slow) based on a median split of RT (within subjects) on correct response". Does this mean that trials for each subject were divided to those with fast and slow RTs or that subjects were divided to fast and slow responders? Please report the median RTs that were used for splitting.

The RT split was done in each subject and experimental condition separately. The resulting subject-selective averages were then averaged to create a grand average (over subjects). The latter served to estimate the source waves and CDR map in Figure 4B. We added text in the results section to clarify this issue. As the median-split differs for each subject, we decided to report the average (over subjects) RT difference between fast and slow response trials.